# Reliability of Repeated Isometric Neck Strength in Rugby Union Players Using a Load Cell Device

**DOI:** 10.3390/s22082872

**Published:** 2022-04-08

**Authors:** Christian Chavarro-Nieto, Martyn Beaven, Nicholas Gill, Kim Hébert-Losier

**Affiliations:** 1Division of Health, Engineering, Computing and Science, Te Huataki Waiora School of Health, University of Waikato, Adams Centre for High Performance, Tauranga 3116, New Zealand; martyn.beaven@waikato.ac.nz (M.B.); nicholas.gill@nzrugby.co.nz (N.G.); kim.hebert-losier@waikato.ac.nz (K.H.-L.); 2New Zealand Rugby, Wellington 6011, New Zealand

**Keywords:** concussion, football, muscle testing, stability, test-retest

## Abstract

Concussion is the most common injury in professional Rugby Union (RU) players, with increasing incidence and severity each year. Strengthening the neck is an intervention used to decrease concussion incidence and severity, which can only be proven effective if strength neck measures are reliable. We conducted a repeated-measures reliability study with 23 male RU players. Neck strength was assessed seated in a ‘make’ test fashion in flexion, extension, and bilateral-side flexion. Flexion-to-extension and left-to-right side ratios were also computed. Three testing sessions were undertaken over three consecutive weeks. Intrasession and intersession reliabilities were assessed using typical errors, coefficient of variations (CV), and intraclass correlation coefficients (ICC). Intrasession reliability demonstrated good-to-excellent relative (ICC > 0.75) and good absolute (CV ≤ 20%) reliability in all directions (ICC = 0.86–0.95, CV = 6.4–8.8%), whereas intersession reliability showed fair relative (ICC: 0.40 to 0.75) and acceptable absolute (CV ≤ 20%) reliability for mean and maximal values (ICC = 0.51–0.69, CV = 14.5–19.8%). Intrasession reliability for flexion-to-extension ratio was good (relative, ICC = 0.86) and acceptable (absolute, CV = 11.5%) and was fair (relative, ICC = 0.75) and acceptable (absolute, CV = 11.5%) for left-to-right ratio. Intersession ratios from mean and maximal values were fair (relative, ICC = 0.52–0.55) but not always acceptable (absolute, CV = 16.8–24%). Assessing isometric neck strength with a head harness and a cable with a load cell device seated in semi-professional RU players is feasible and demonstrates good-to-excellent intrasession and fair intersession reliability. We provide data from RU players to inform practice and assist standardisation of testing methods.

## 1. Introduction

Concussion is consistently ranked as the most common injury in professional male English Rugby Union players (21%), with increasing incidence and severity from 2002 to 2019 [1]. Considering the medical impact of concussion in Rugby Union, various strategies are being implemented and trialled to reduce the incidence and severity of these injuries. Given that the tackler is the player with the largest incidence of concussion, World Rugby introduced a series of evidence-based initiatives to lessen concussion rates and severity and improve player welfare [2]. These initiatives include improving the tackling technique, reducing the height of the tackle, and changing the speed of the tackle, which presumably have contributed to the 12% reduction in concussions at the 2019 Rugby World Cup compared to the 2015 tournament [3]. In Rugby Union, the average momentum when tackling another player is above 320 kg·m/s, and this elicits high levels of muscle activation at the neck and shoulder, particularly in previously injured players [4]. While rule changes have been identified as one strategy to address concussion in Rugby Union [3], neck strength has also been identified as a modifiable risk factor [5,6]. Training and return to sport protocols highlight the clinical and rehabilitative value of neck strength in neck and head injuries [7,8]. Furthermore, flexion-to-extension and left-to-right lateral flexion imbalances in neck strength are potential risk factors for neck injuries, wherein flexion-to-extension imbalances have been associated with higher head angular and linear accelerations in other cohorts than Rugby Union [9]. These ratios are often assessed in Rugby Union, with a review summarising greater strength values in extension compared to flexion [10], with forwards possessing a significantly lower ratio (0.56) compared to backs (0.61) [11]. Considering bilateral lateral flexion ratios, left to right ratio values range from 0.98 in backs and 1.00 in forwards across the literature [11].

Currently, variations exist in neck strength assessment methods, benchmark values, and exercise prescription in Rugby Union [10]. The most common test used to assess neck strength in Rugby Union is the ‘break’ test, which measures peak isometric neck strength when resisting against an incrementally applied load [10]. Although the break test is the most common method used in Rugby Union, clinicians have expressed concerns regarding the ‘aggressiveness’ of this method and the potential to injure players during testing [12]. The resistance is applied to evoke a maximal muscle contraction in flexion, extension, and bilateral lateral flexion directions using a head harness attached to a cable and dynamometer or load cell until the initiation of the movement (i.e., positional failure) [11,13,14,15,16,17]. Previous reliability studies with this neck strength assessment method demonstrate good-to-excellent intrarater and interrater reliability (intraclass correlation coefficient (ICC) 0.80–0.92) when implemented in seated or lying positions [11,18].

An alternative to the ‘break’ test is to use the ‘make’ test, which is performed using a head harness attached to a load cell device to a fixed frame or tested against a manual resistance in seated or lying positions with the participant exerting maximal strength against the load cell or handheld dynamometer [7,19,20]. The reliability of make test isometric neck strength measures from a load cell in healthy individuals was excellent between-day (ICC 0.94–0.98) performed in sitting position [21]. In semi-professional Rugby Union players, the make test intrasession reliability assessed using a fixed-mounted handheld dynamometer was good for flexion, extension, and bilateral lateral flexion (ICC 0.77–0.92), as well as when tested against a manual resistance in these four directions (ICC 0.77–0.90) [22]. However, intersession reliability for the neck test in Rugby Union has not been examined. The lack of data on the between-day reliability of the make test in Rugby Union might explain research showing no significant changes in make test neck strength measures during a 26-week neck strengthening programme in Rugby Union despite a significant decrease in neck match-injuries [7].

It is of upmost importance that clinicians use reliable methods when assessing individuals to inform their clinical practice. It is also essential that normative data for neck strength values be available for male Rugby Union players to inform management of this unique cohort, particularly in rehabilitation and return to play. In Rugby Union, there are no studies examining the reliability of an isometric make test using a rigid cable with load cell in a seated position set-up or information on the between day reliability on the make test. Therefore, we aimed to examine the reliability of make test isometric neck strength measures in Rugby Union players using a load cell device.

## 2. Materials and Methods

### 2.1. Study Design

A repeated-measures reliability study was conducted targeting semi-professional male Rugby Union players. Based on methods described to establish minimum sample size requirements for reliability studies [23], a minimum of around 20 participants was needed when setting the acceptable reliability level at ρ0 = 0.40 (i.e., fair reliability threshold) and desired reliability level at ρ1 > 0.75 (i.e., good reliability threshold) with an α = 0.05 and β = 0.20 knowing that players were assessed on three occasions.

### 2.2. Participants

Twenty-three semi-professional male Rugby Union players (mean ± standard deviation (SD), age 22.3 ± 3.2 year, height 184 ± 7.5 cm, and body mass 100 ± 11 kg) agreed to participate in this study that took place during their regular season. Inclusion criteria required participants to be free of neck pain, cervical injury, or illness in the last month as to not compromise maximal isometric neck strength. All participants were informed of the purpose, benefits, and risks of participation through written and oral description, and they gave their written informed consent to participate. The study protocol was approved by the University of Waikato Human Research Ethics Committee (HREC(Health) 2019#74) and adhered to the latest Declaration of Helsinki.

### 2.3. Instrumentation

Tests were conducted using a SOUFEI digital portable load cell (SF-912 Soufei Electronic Technology Co., Ltd., Jiangyin, China) of 100 g accuracy with the capacity set to 50 kg. The load cell was connected via Bluetooth to an iPad Model A1566 device, and data were recorded at 20 Hz. Participants were seated in a standard fixed gym seat bench for testing.

### 2.4. Procedures

This study assessed the neck strength in a make test fashion in four different directions: flexion, extension, and right and left lateral flexion. Each player completed three testing sessions over three consecutive weeks, one week apart. Before the experimental procedure, all participants completed a warm-up protocol of three submaximal isometric repetitions in each direction tested.

For the experimental procedure, we followed procedures similar to those reported elsewhere [24] and implemented in professional Rugby Union players [7]. Participants were seated on the standard gym bench with the head and neck in a neutral position. A seat belt secured the trunk to minimise movement, and two balance air pads were placed under the feet to prevent further movement and contributions from the lower limbs. A head harness was placed around the forehead for flexion, occiput for extension, and temporal bones for lateral flexion. The harness was attached to a load cell apparatus via a rigid cable attached to a fixed metal frame (Figure 1). For data collection, participants completed three maximal effort repetitions in each of the four directions tested, with 30-s rest between contractions, and 60-s rest between directions. The order of testing was randomized between participants and held constant between testing occasions. The instructions provided before the make test was to “pull as hard as you can with 100% strength and hold for 5 s”. The peak strength in kilograms was recorded for each isometric contraction and used as the main outcome measure. 

### 2.5. Statistical Analysis

Data are described using means ± SD. The normal distribution of variables was assessed with Shapiro–Wilks’s and d’Agostino–Pearson tests. Data were log-transformed for reliability analysis to reduce bias arising from non-uniformity of error when appropriate. The three repetitions completed during the first session were used to examine the intrasession reliability for each direction. For intersession reliability, analysis of both mean and maximal values was undertaken for each direction. The intersession reliability reflects the stability of measures as it defines the day-to-day variability in measures, which typically needs more than one-day between measures in sport measures [25]. The reliability of intrasession and intersession measurements was assessed using intraclass correlation coefficient (ICC), coefficient of variation (CV), typical error (TE), and mean change (Δ), and these were calculated with their SD or 95% confidence limits [lower, upper] using a customised statistical Excel spreadsheets [26] in Microsoft Excel for Office MSO (Version 2111 Build 16.0.14701.20254). Relative reliability was interpreted as poor, fair, good, and excellent when corresponding ICCs were <0.40, 0.40 to 0.75, >0.75 to 0.90, and >0.90 [27]. Absolute reliability was considered good and acceptable when corresponding CVs were ≤10% and ≤20% [28,29].

Trials and repetitions were assessed for systematic error (i.e., learning effects) using a one-way repeated measures analysis of variance (RM ANOVA) using STATA. The Duncan method was applied in post hoc testing. Statistical significance level was set at *p* ≤ 0.05 for all analysis. If the assumption of sphericity was violated, the adjusted *p*-values were reported. 

## 3. Results

Descriptive and reliability statistics related to intrasession isometric neck strength values and ratios are shown in Table 1. Those related to intersession mean values are displayed in Table 2, and intersession maximal values are reported in Table 3.

### 3.1. Extension

Isometric neck extension demonstrated good intrasession reliability (ICC = 0.86, TE = 2.5 kg, and CV = 8.8%), fair intersession reliability for mean values (ICC = 0.51, TE = 4.5 kg, and CV = 15.9%), and fair intersession reliability for maximal values (ICC = 0.54, TE = 4.9 kg, and CV = 15.1%). There was no systematic bias across reliability analyses based on the RM ANOVAs (*p* ≥ 0.732).

### 3.2. Flexion

Isometric neck flexion demonstrated excellent intrasession reliability (ICC = 0.91, TE = 1.6 kg, and CV = 6.7%), fair intersession reliability for mean values (ICC = 0.60, TE = 3.3 kg, and CV = 14.5%), and fair intersession reliability for maximal values (ICC = 0.58, TE = 3.3 kg, and CV = 14.5%). There was no systematic bias across reliability analyses based on the RM ANOVAs (*p* ≥ 0.053).

### 3.3. Left Lateral Flexion

Isometric neck left lateral flexion demonstrated excellent intrasession reliability (ICC = 0.95, TE = 1.6 kg, and CV = 6.4%), fair intersession reliability for mean values (ICC = 0.56, TE = 3.9 kg, and CV = 18.1), and fair intersession reliability for maximal values (ICC = 0.63, TE = 4.0 kg, and CV = 18.6%). There was no systematic bias for intrasession and mean intersession analyses based on the RM ANOVAs (*p* ≥ 0.058). However, bias was detected for intersession reliability analysis of maximal values (*p* = 0.044). Post hoc testing revealed a significant difference between Trial 3 and 1 (*p* = 0.031), and Trial 3 and 2 (*p* = 0.033), with lower values at the third session. 

### 3.4. Right Lateral Flexion

Isometric neck right lateral flexion demonstrated excellent intrasession reliability (ICC = 0.88, TE = 1.9 kg, and CV = 7.9%), fair intersession reliability for mean values (ICC = 0.53, TE = 5.0 kg, and CV = 19.8%), and fair intersession reliability for maximal values (ICC = 0.59, TE = 5.0 kg, and CV = 19.8%). There was no systematic bias across reliability analyses based on the RM ANOVAs (*p* ≥ 0.431).

### 3.5. Flexion-to-Extension and Lateral Left-to-Right Ratios

Intrasession flexion-to-extension ratio values ranged from 0.77 to 0.81 with excellent reliability (ICC = 0.86, TE = 0.13, and CV = 11.5%). Intrasession lateral left-to-right ratio values ranged from 0.97 to 0.99 with fair reliability (ICC = 0.75, TE = 0.08, and CV = 8.2%). Intersession mean flexion-to-extension ratios demonstrated fair reliability (ICC = 0.55, TE = 0.82, and CV = 24%) as did lateral left-to-right ratios (ICC = 0.52, TE = 0.94, and CV = 18.8%). Intersession reliability for maximal flexion-to-extension (ICC = 0.53, TE = 0.77, and CV = 20.7%) and lateral left-to-right (ICC = 0.53, TE = 0.89, and CV: 16.8%) ratios were fair. There was no systematic bias across reliability analyses based on the RM ANOVAs (*p* ≥ 0.649).

## 4. Discussion

We examined the reliability of make test isometric neck strength measures in male Rugby Union players using a load cell device in extension, flexion, and left and right lateral flexion. Reliability of flexion-to-extension and left-to-right lateral flexion ratios were also examined. Intrasession reliability was good to excellent (ICC = 0.85–0.95), whereas intersession reliability–which here reflects the stability in measures–was fair for both mean and maximal values (ICC = 0.51–0.69). Similar make test isometric methods have been used to assess Rugby Union players, but reliability was not assessed [7], providing new insight, particularly with regards to intersession reliability and stability of measures.

Our intrasession reliability results coincide with previous studies in semi-professional Rugby Union players using different devices, wherein intrasession reliability of make test isometric neck strength measures using a fixed-mounted handheld dynamometer was good to excellent for extension, flexion, and bilateral lateral flexion (ICC = 0.77–0.92), as well as when tested against a manual resistance in these four directions (ICC = 0.77–0.90) [22]. When the reliability was assessed in a break test fashion with a handheld dynamometer in a seated position in academy Rugby Union players, intrasession reliability was good but did not reach excellent across directions (ICC = 0.80–0.85) [11]. Regarding values of absolute reliability, intrasession reliability was good with CV ≤ 10%, and intersession reliability for mean and maximal values were acceptable with CV ≤ 20%. When tested using a computerized dynamometer in a seated position, no significant intrasession or intersession differences were detected in non-Rugby Union cohorts, with strong correlations between measures for intrasession (r 0.951–0.989) and intersession (r 0.731–0.969). The CV were <15% when assessed in a neutral neck position and reached 26% in 10° of neck extension, suggesting that both intrasession and intersession assessments were reliable [30]. Our results indicate that the current method to test neck strength in Rugby Union within the same session is good, and we here provide new information regarding intersession reliability and stability of measures. When tested a week apart, reliability can be considered acceptable. The intersession reliability was the weakest for extension, which aligns with findings on flexion and extension intersession isometric neck strength assessed using a scrum device in university school Rugby Union players [31]. The lower reliability in extension was thought to be due to variations in technique and body positioning between sessions [31]. It would seem that using a head harness and cable with a load cell device in a seated position in Rugby Union players is also subject to variations in position between sessions, which could explain the superior intrasession than intersession reliability outcomes. 

To our knowledge, this is the first study examining the reliability of an isometric neck strength test in a make test manner using a rigid cable, a head harness with load cell device, in a seated position in Rugby Union players. Our study found neck strength values in semi-professional Rugby Union players range from 22 to 33 kg in all directions. These maximal isometric neck strength values are similar to those observed with amateur players assessed in a contact position using a load cell or fixed-frame dynamometer [32]. Our ranges were, however, greater than those from isometric neck strength values assessed in a cohort of semi-professional Rugby Union players tested using a mounted handheld dynamometer and a make test approach (range 17.3–23.5 kg) [22]. The difference potentially reflects the superior procedural methods of the load cell with harness and cable set-up than the handheld dynamometer set-up due to the testing interaction between the participant and examiner in the latter method [22]. When professional players were assessed in a similar fashion to ours, specifically in a seated position with a customized load cell device, a head harness, and an immovable metal frame in a make test fashion, the maximal strength values were greater than those reported here (i.e., 29–39 kg) [7]. Several studies have attempted to identify normative neck strength values in Rugby Union players with different methods of testing [10]; however, there is no agreement on the strength testing method to be used or requisite strength levels [10]. The results of the present study provide data on isometric neck strength for semi-professional male Rugby Union players in flexion, extension, and left and right lateral flexion, as well as flexion-to-extension and left-to-right lateral flexion ratios using a reproducible set-up. In addition, our study indicates that reliability is good-to-excellent intrasession and fair intersession when players are tested in-season. 

Extension-to-flexion ratios ranged from 0.74 and 0.82 with good intrasession reliability, and left-to-right lateral flexion ratios were around 0.97 to 1.01 with fair intrasession reliability. Intersession reliability was fair for both ratios. Our results are similar to those reported elsewhere in semi-professional Rugby Union players with flexion-to-extension ratios of 0.75 to 0.76 and bilateral lateral flexion ratios of 0.96 to 1.0 in a make test fashion with a load cell device via cable in a seated position [7,11]. Noteworthy is that these ratios are marginally greater than those presented elsewhere using different methods of testing (isokinetic and break test), wherein flexion-to-extension ratios in Rugby Union ranged, on average, from 0.65 to 0.7 [14,33]. Results of previous studies at different levels of competition have shown greater absolute values in extension compared to flexion in Rugby Union players [34,35,36], with younger players possessing lower strength ratios compared to senior players [17], and forwards exhibiting lower ratios than backs [19,33]. These imbalances in neck musculature are a potential risk factor for neck injuries, where flexion-to-extension ratio imbalances have been associated with higher head angular and linear accelerations in other cohorts [37]. Promoting strength symmetry between muscles is a strategy used in practice to mitigate injury risk in other anatomical parts of the body (e.g., knee) [38] and should be explored further as a mean to mitigate concussion and neck injury risk in Rugby Union.

## 5. Conclusions

Assessing isometric neck strength with a head harness and a cable with a load cell device in a seated position in semi-professional Rugby Union players is feasible and demonstrates good-to-excellent intrasession and fair intersession relative reliability. Absolute reliability is good intrasession (CV ≤ 10%) and acceptable intersession (CV ≤ 20%). Although the load cell approach appears less reliable than using a handheld dynamometer based on reliability values reported in the literature, the load cell approach can be reliably used to assess isometric neck strength in extension, flexion, and bilateral lateral flexion, as well as to derive flexion-to-extension and left-to-right lateral flexion ratios in male Rugby Union players. A direct comparison of reliability data from the same cohort using both the handheld dynamometer and cable load cell approach is needed to confirm that one method is more reliable than the other. We provide data from Rugby Union players to inform practice and assist standardisation of testing methods. Further experimentation with the load cell device to improve intersession reliability and stability of measures in neck isometric testing is recommended. Intersession reliability might be improved using three-axial load cells, ensuring an initial familiarisation session and further attention to individualised and repeatable neck positions.

## Figures and Tables

**Figure 1 sensors-22-02872-f001:**
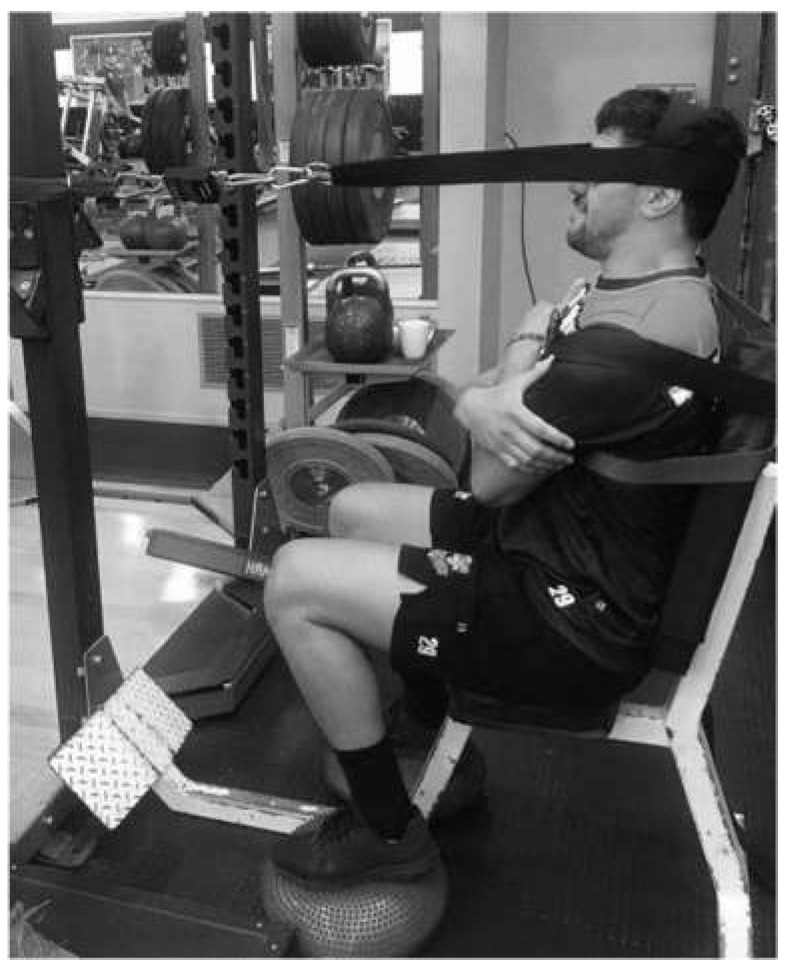
Illustration of the isometric make neck strength test experimental setup for extension.

**Table 1 sensors-22-02872-t001:** Descriptive and reliability statistics related to intrasession isometric neck strength values. Values include mean, standard deviation, and 95% confidence intervals [upper, lower] for the four different directions examined.

	Isometric Strength (Raw Units)			Δ Isometric Strength (Raw Units)			Reliability Statistics			
	T1	T2	T3	T1–T2	T2–T3	T1–T3	ICC [95% CI]	TE (kg) [95% CI]	CV (%) [95% CI]	*p*-Value ^a^
**Extension (kg)**	31.9 (5.6)	32.3 (6.4)	31.4 (7.5)	0.5 (3.4)	−0.9 (3.4)	−0.5 (3.4)	0.86 [0.75–0.93]	2.5 [2.0–3.0]	8.8 [7.1–11.7]	0.538
**Flexion (kg)**	23.9 (4.7)	25.8 (5.9)	25.7 (5.6)	0.2 (2.1)	0.1 (2.5)	−0.3 (2.0)	0.91 [0.84–0.95]	1.6 [1.3–1.9]	6.7 [5.6–8.5]	0.809
**Left lateral flexion (kg)**	25.4 (5.9)	25.6 (6.4)	25.6 (6.3)	0.2 (2.1)	−0.1 (2.5)	−0.2 (2.1)	0.95 [0.90–0.97]	1.6 [1.4–1.9]	6.4 [5.6–8.5]	0.873
**Right lateral flexion (kg)**	26.1 (5.3)	25.8 (5.9)	25.7 (5.6)	−0.4 (2.0)	−0.1 (2.7)	0.4 (3.3)	0.88 [0.78–0.94]	1.9 [1.7–2.3]	7.9 [6.5–10.7]	0.734
**Flexion/extension** **ratio**	0.77 (0.20)	0.77 (0.21)	0.82 (0.31)	0.0 (0.3)	0.1 (0.2)	0.1 (0.3)	0.86 [0.69–0.91]	0.13 [0.11–0.15]	11.5 [9.6–11.5]	0.363
**Left/right** **ratio**	0.98 (0.15)	1.00 (0.13)	1.00 (0.16)	0.1 (0.2)	0.0 (0.3)	0.1 (0.2)	0.75 [0.58–0.87]	0.08 [0.07–0.10]	8.2 [6.9–10.7]	0.617

**^a^** *p*-value from repeated measures analysis of variance. CI: Confidence interval, CV: coefficient of variation, ICC: intraclass correlation coefficient, SD: standard deviation, T1: trial 1, T2: trial 2, T3, trial 3, TE: Typical error.

**Table 2 sensors-22-02872-t002:** Descriptive and reliability statistics related to intersession isometric neck strength (mean of three trials). Values include mean, standard deviation, and 95% confidence intervals [upper, lower] for the four different directions examined.

	Isometric Strength (Raw Units)			Δ Isometric Strength (Raw Units)			Reliability Statistics			
	T1	T2	T3	T1–T2	T2–T3	T1–T3	ICC [95% CI]	TE (kg) [95% CI]	CV (%) [95% CI]	*p*-Value ^a^
**Extension (kg)**	32.0 (6.0)	31.7 (6.4)	31.2 (6.7)	0.1 (8.2)	−0.4 (3.2)	0.7 (8.4)	0.51 [0.26–0.73]	4.5 [3.6–6.3]	15.9 [12.6–22.6]	0.732
**Flexion (kg)**	24.0 (4.8)	23.5 (5.7)	22.1 (4.3)	−0.7 (4.3)	−2.5 (4.9)	3.1 (5.1)	0.60 [0.35–0.77]	3.3 [2.8–4.1]	14.5 [12.0–19.5]	0.053
**Left lateral flexion (kg)**	25.5 (6.1)	24.9 (4.1)	22.8 (6.4)	0.1 (5.5)	−2.0 (5.3)	2.1 (6.1)	0.56 [0.32–0.75]	3.9 [3.4–5.0]	18.1 [14.9–24.4]	0.058
**Right lateral flexion (kg)**	25.7 (5.5)	25.6 (6.6)	24.2 (8.2)	0.6 (8.0)	−1.4 (4.6)	1.1 (8.2)	0.53 [0.27–0.73]	5.0 [4.3–6.3]	19.8 [16.3–26.9]	0.431
**Flexion/extension** **ratio**	0.78 (0.27)	0.77 (0.28)	0.74 (0.13)	0.0 (0.3)	−0.1 (0.2)	0.1 (0.3)	0.55 [0.29–0.74]	0.82 [0.70–1.02]	24.0 [19.7–32.7]	0.649
**Left/right** **ratio**	1.00 (0.27)	1.01 (0.28)	0.96 (1.37)	0.0 (0.3)	−0.1 (0.2)	−0.1 (0.2)	0.52 [0.23–0.76]	0.94 [0.8–1.16]	18.8 [0.2–0.8]	0.646

**^a^** *p*-value from repeated measures analysis of variance. CI: Confidence interval, CV: coefficient of variation, ICC: intraclass correlation coefficient, SD: standard deviation, T1: trial 1, T2: trial 2, T3, trial 3, TE: Typical error.

**Table 3 sensors-22-02872-t003:** Descriptive and reliability statistics related to intersession isometric neck strength (maximal value from three trials). Values include mean, standard deviation, and 95% confidence intervals [upper, lower] for the four different directions examined.

	Isometric Strength (Raw Units)			Δ Isometric Strength (Raw Units)			Reliability Statistics			
	T1	T2	T3	T1–T2	T2–T3	T1–T3	ICC [95% CI]	TE (kg) [95% CI]	CV (%) [95% CI]	*p*-Value ^a^
**Extension (kg)**	33.6 (6.3)	33.4 (6.2)	33.1 (6.8)	0.4 (8.1)	−0.1 (3.6)	0 (8.4)	0.54 [0.25–0.76]	4.9 [4.0–6.2]	15.1 [12.0–21.5]	0.849
**Flexion (kg)**	25.3 (5.2)	25.1 (5.9)	23.4 (4.7)	-0.5 (4.6)	−2.8 (5.4)	3.3 (5.4)	0.58 [0.33–0.76]	3.6 [3.0–4.8]	14.5 [12.0–19.5]	0.055
**Left lateral flexion (kg)**	26.8 (6.3)	26.8 (4.7)	24.2 (6.4)	0.4 (5.6)	−2.4 (5.6)	1.6 (5.9)	0.69 [0.43–0.85]	4.0 [3.4–4.8]	18.6 [14.9–26.0]	0.044 *
**Right lateral flexion (kg)**	27.6 (5.4)	26.0 (4.5)	25.0 (7.1)	-0.9 (5.1)	−0.8 (5.5)	2.4 (4.9)	0.60 [0.35–0.77]	3.6 [3.12–4.53]	16.9 [13.9–22.8]	0.097
**Flexion/extension** **ratio**	0.74 (0.21)	0.73 (0.25)	0.70 (0.20)	0.0 (0.3)	−0.1 (0.2)	0.1 (0.2)	0.53 [0.27–0.77]	0.77 [0.66–0.95]	20.7 [17.0–28.1]	0.547
**Left/right** **ratio**	0.97 (0.14)	1.04 (0.16)	0.97 (0.11)	0.1 (0.2)	−0.1 (0.2)	0.0 (0.2)	0.53 [0.26–0.73]	0.89 [0.76–1.08]	16.8 [13.5–24.0]	0.162

* *p* < 0.05. ^a^
*p*-value from repeated measures analysis of variance. CI: Confidence interval, CV: coefficient of variation, ICC: intraclass correlation coefficient, SD: standard deviation, change, T1: trial 1, T2: trial 2, T3, trial 3, TE: Typical error. Repeated measures analysis of variance significance set at *p* < 0.05.

## Data Availability

Not applicable.

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
