# Peer review of "Reliability of Repeated Isometric Neck Strength in Rugby Union Players Using a Load Cell Device"

_sensors, 2022, doi:10.3390/s22082872_

Round 1

Reviewer 1 Report

The authors have conducted a solid study that addresses a relevant question. The practical focus of the work is well explained and it has clear relevance for future research and practice. I enjoyed reading the article. 

I have one major concern but it is one that I hope the authors appreciate is designed to enhance the paper. The data analysis could do with examining the stability of data - do we get the same or very similar score in the re-retest condition when we expect stability. Researchers have questioned the use of correlation methods and argued that reliability could be high but stability could be low. The work by Atkinson and Nevill (1998) began this discussion. I encourage the authors to have a discussion on stability and reliability in their paper, do an up-to-date search on the literature to consider any recent developments and then either re-analyse their data (which I think they should do as the test-retest agreement methods provide simple answers of agreement) or argue why they don't need to (which I think would be difficult to do well and not look like they were selective in the methods used). 

Statistical methods for assessing measurement error (reliability) in variables relevant to sports medicine G Atkinson, AM Nevill - Sports medicine, 1998 Cited by 3735 Related articles All 12 versions   On doing the extra data analysis, we will get some new results. These might not change the results and if so, the discussion remains good. A paragraph could be added on the value of having multiple methods to assess reliability/stability in the search to produce rigour in research.    In summary, an interesting paper that has real potential to have practical impact. I would like to see the rigour in the results beefed up a little.

Author Response

We would like to thank the editorial team and peer-reviewers for their critical appraisal and feedback on our manuscript. We are pleased with the invitation to resubmit our work to Sensors for publication consideration pending satisfactory revisions and responses to the reviewers’ comments.

As requested, we have revised our manuscript according to the comments received and provide below point-by-point answers to all comments. To facilitate the review process, we have indicated all modifications in the manuscript in RED font colour, except for areas where we removed text. We trust that the reviewers will find our responses meet their expectations. The feedback provided was helpful for improving the quality of our manuscript.

Reviewer: 1

Q1: Comments to the Author
The authors have conducted a solid study that addresses a relevant question. The practical focus of the work is well explained and it has clear relevance for future research and practice. I enjoyed reading the article. 

I have one major concern but it is one that I hope the authors appreciate is designed to enhance the paper. The data analysis could do with examining the stability of data - do we get the same or very similar score in the re-retest condition when we expect stability. Researchers have questioned the use of correlation methods and argued that reliability could be high but stability could be low. The work by Atkinson and Nevill (1998) began this discussion. I encourage the authors to have a discussion on stability and reliability in their paper, do an up-to-date search on the literature to consider any recent developments and then either re-analyse their data (which I think they should do as the test-retest agreement methods provide simple answers of agreement) or argue why they don't need to (which I think would be difficult to do well and not look like they were selective in the methods used). 

Statistical methods for assessing measurement error (reliability) in variables relevant to sports medicine G Atkinson, AM Nevill - Sports medicine, 1998 Cited by 3735 Related articles All 12 versions   On doing the extra data analysis, we will get some new results. These might not change the results and if so, the discussion remains good. A paragraph could be added on the value of having multiple methods to assess reliability/stability in the search to produce rigour in research.    In summary, an interesting paper that has real potential to have practical impact. I would like to see the rigour in the results beefed up a little.

We thank the reviewer for his/her comments.

As per Atkinson and Nevill (1998): “Stability reliability was defined as the day-to-day variability in measurements. This is the most common type of reliability analysis, although

it is stressed that exercise performance tests may need more than one day between repeated measurements to allow for bias due to inadequate recovery. Hence, our intersession reliability data can be understood to reflect stability in measures as was tested on three occasions, seven days apart (as stated in the 2.4 Procedures section): “Each player completed three testing sessions over three consecutive weeks, one week apart”. To highlight this more clearly, we have added the following statement in the methods section:

2.5 Statistical analysis: The intersession reliability reflects the stability of measures as it defines the day-to-day variability in measures, which typically needs more than one-day between measures in sport measures [1].

To highlight this point further, we have made the following clarifications in the Discussion and Conclusions.

Discussion: Intrasession reliability was good to excellent (ICC = 0.85–0.95), whereas intersession reliability – which here reflects the stability in measures – was fair for both mean and maximal values (ICC = 0.51–0.69).

Discussion: Similar make test isometric methods have been used to assess Rugby Union players, but reliability was not assessed [6], providing new insight, particularly with regards to intersession reliability and stability of measures.

Conclusions: Further experimentation with the load cell device to improve intersession reliability and stability of measures in neck isometric testing is recommended.

Given that ICCs should not be interpreted in isolation (relative reliability), we also report data on absolute reliability (typical error and coefficient of variation), which highlights the amount of error. Systematic bias as also examined via ANOVA. Hence, the authors feel we have covered the spectrum of reliability statistics appropriate for this data set.

Reviewer 2 Report

The article “Reliability of Repeated Isometric Neck Strength in Rugby Union Players Using a Load Cell Device“ is novel and provides good pilot information about neck Strength testing by easily accessible devices. However, the conclusion of the article is not done with a sufficient overview and unbiased recommendations. Thus, the main part to improve this article is to change interpretations in intersection reliability. Authors might suggest investigating by three-axial load cells, to get desired reliability. Perhaps better testing protocol.

The methods are replicable and the result clearly presented. Discussion critically compares the results of the study to previous findings, however, the conclusion ignores that handheld dynamometer has better reliability. The CV 15-20% with ICC 0.53-0.60 means very questionable reliability. Therefore, I strongly suggest making different conclusions out of the results. E.g. recommending to make normative data with such questionable reliability is not relevant.

Line 9, 12: Why there are brackets with numbers in general abstract design?

Line 23: Statement that intersection reliability was fair should have a number in bracket, what the fair means. Intersection variability of about 20% is too high.

Line 25 26: some keywords are already in the title, use more general terms.

Line 36: What are the energy or moments during a tackling?

Line 49 – 73: Isometric strength seems to be fair, what is its disadvantage or impractical? This stating is in the following paragraph but should be here. Moreover, your device is also isometric.

Line 141: The reference for reliability is not methodologically based and there is only reported Doi form 1999 y. Use more strong reference, which has full data.

Line 215 – 219: This discussion clearly shows that your device is less reliable than handheld dynamometry, this should be part of the conclusion.

Line 282: fair intersession reliability actually means that better methods (more reliable) should be used for normative values. The conclusion should state whether your method is more or less reliable than others.  

Author Response

We would like to thank the editorial team and peer-reviewers for their critical appraisal and feedback on our manuscript. We are pleased with the invitation to resubmit our work to Sensors for publication consideration pending satisfactory revisions and responses to the reviewers’ comments.

As requested, we have revised our manuscript according to the comments received and provide below point-by-point answers to all comments. To facilitate the review process, we have indicated all modifications in the manuscript in RED font colour, except for areas where we removed text. We trust that the reviewers will find our responses meet their expectations. The feedback provided was helpful for improving the quality of our manuscript.

Reviewer: 2

Q1: Comments to the Author

The article “Reliability of Repeated Isometric Neck Strength in Rugby Union Players Using a Load Cell Device“ is novel and provides good pilot information about neck Strength testing by easily accessible devices. However, the conclusion of the article is not done with a sufficient overview and unbiased recommendations. Thus, the main part to improve this article is to change interpretations in intersection reliability. Authors might suggest investigating by three-axial load cells, to get desired reliability. Perhaps better testing protocol.

The methods are replicable and the result clearly presented. Discussion critically compares the results of the study to previous findings, however, the conclusion ignores that handheld dynamometer has better reliability. The CV 15-20% with ICC 0.53-0.60 means very questionable reliability. Therefore, I strongly suggest making different conclusions out of the results. E.g. recommending to make normative data with such questionable reliability is not relevant.

Q1: Comments to the Author

The article “Reliability of Repeated Isometric Neck Strength in Rugby Union Players Using a Load Cell Device“ is novel and provides good pilot information about neck Strength testing by easily accessible devices. However, the conclusion of the article is not done with a sufficient overview and unbiased recommendations. Thus, the main part to improve this article is to change interpretations in intersection reliability. Authors might suggest investigating by three-axial load cells, to get desired reliability. Perhaps better testing protocol.

The methods are replicable and the result clearly presented. Discussion critically compares the results of the study to previous findings, however, the conclusion ignores that handheld dynamometer has better reliability. The CV 15-20% with ICC 0.53-0.60 means very questionable reliability. Therefore, I strongly suggest making different conclusions out of the results. E.g. recommending to make normative data with such questionable reliability is not relevant.

R1. Thank you for your comments. We have altered the conclusions to align with your comments, while respecting the thresholds that we highlighted in our statistical procedures section. It is also worth noting that to state that handheld dynamometer methods are better in terms of reliability than what we report here for the load cell device would need a study performed with the same cohort of athletes, directly comparing the two methods. At this point in time, the superior reliability of handheld can only be inferred from the available literature, which does not consider the biological variations (for instance) of different cohorts.

Conclusions: Assessing isometric neck strength with a head harness and a cable with a load cell device in a seated position in semi-professional Rugby Union players is feasible, and demonstrates good-to-excellent intrasession and fair intersession relative reliability. Absolute reliability is good intrasession (CV ≤ 10%) and acceptable intersession (CV ≤ 20%). Although the load cell approach appears less reliable than using a handheld dynamometer based on reliability values reported in the literature, the load cell approach can be reliably used to assess isometric neck strength in extension, flexion, and bilateral lateral flexion, as well as to derive flexion-to-extension and left-to-right lateral flexion ratios in male Rugby Union players. A direct comparison of reliability data from the same cohort using both the handheld dynamometer and cable load cell approach is needed to confirm that one method is more reliable than the other. We provide data from Rugby Union players to inform practice, and assist standardisation of testing methods. Further experimentation with the load cell device to improve intersession reliability and stability of measures in neck isometric testing is recommended. Intersession reliability might be improved using three-axial load cells, ensuring an initial familiarisation session, and further attention to individualised and repeatable neck positions. 

Q2: Abstract:
Line 9, 12: Why there are brackets with numbers in general abstract design?

R2: We have removed the brackets from the abstract.

Q3 Line 23: Statement that intersection reliability was fair should have a number in bracket, what the fair means. Intersection variability of about 20% is too high.

R3: We have included in parentheses the associated ranges, and clarified whether we were referring to absolute or relative reliability throughout the abstract. See changes implemented below:

Abstract:  Intrasession reliability demonstrated good-to-excellent relative (ICC > 0.75) and good absolute (CV ≤ 20%) reliability in all directions (ICC=0.85–0.95, CV=6.4–8.8%), whereas intersession reliability showed fair relative (ICC: 0.40 to 0.75) and acceptable absolute (CV ≤ 20%) reliability for mean and maximal values (ICC=0.51–0.69, CV=14.5–19.8%). Intrasession reliability for flexion-to-extension ratio was good (relative, ICC=0.86) and acceptable (absolute, CV=11.5%), and was fair (relative, ICC=0.75) and acceptable (absolute, CV=11.5%) for left-to-right ratio. Intersession ratios from mean and maximal values were fair (relative, ICC=0.52–0.55), but not always acceptable (absolute, CV=16.8–24%).

We clearly define our thresholds in the 2.5 Statistical analysis section with associated references.

2.5 Statistical analysis: Relative reliability was interpreted as poor, fair, good, and excellent when corresponding ICCs were < 0.40, 0.40 to 0.75, > 0.75 to 0.90, and > 0.90 [24]. Absolute reliability was considered good and acceptable when corresponding CVs were ≤ 10% and ≤ 20% [25, 26].

Q4. Line 25 26: some keywords are already in the title, use more general terms.

R4: We have used more general terms:  

Keywords: concussion, football, muscle testing, stability, test-retest. 

Q5. Line 36: What are the energy or moments during a tackling?

R5. Based on the supplied reference, we have added the following information.

Introduction: In Rugby Union, the average momentum when tackling another player is above 320 kg∙m/s, and elicits high levels of muscle activation at the neck and shoulder, particularly in previously injured players [3].

Q6. Line 49 – 73: Isometric strength seems to be fair, what is its disadvantage or impractical? This stating is in the following paragraph but should be here. Moreover, your device is also isometric.

R6. We have moved the statement to the prior paragraph as follow:

“Although the break test is the most common method used in Rugby Union, clinicians have expressed concerns regarding the ‘aggressively’ of this method and potential to injure players during testing [17].”

Q7. Line 141: The reference for reliability is not methodologically based and there is only reported Doi form 1999 y. Use more strong reference, which has full data.

R7. We have updated the reference as requested

[4]  Peolsson A, Oberg B, Hedlund R. Intra- and inter-tester reliability and reference values for isometric neck strength. Physiother Res Int. 2001;6(1):15-26. doi: 10.1002/pri.210.

Q8. Line 215 – 219: This discussion clearly shows that your device is less reliable than handheld dynamometry, this should be part of the conclusion.

R8. We have restated:

Conclusion: Although the load cell approach appears less reliable than using a handheld dynamometer based on reliability values reported in the literature, the load cell approach can be reliably used to assess isometric neck strength in extension, flexion, and bilateral lateral flexion, as well as to derive flexion-to-extension and left-to-right lateral flexion ratios in male Rugby Union players. A direct comparison of reliability data from the same cohort using both the handheld dynamometer and cable load cell approach is needed to confirm that one method is more reliable than the other.

General comment: The authors have conducted a solid study that addresses a relevant question. The practical focus of the work is well explained and it has clear relevance for future research and practice. I enjoyed reading the article. 

Round 2

Reviewer 1 Report

The authors  have done an excellent job addressing my comments. I feel its a better paper. 

Reviewer 2 Report

The authors made a great job in their responses and conclusion improvement. Therefore I recommend this article be published.